# A Qualitative Analysis of Management Perspectives on Seeking to Implement the Foster Cat Project in Residential Aged Care in the Context of COVID-19

**DOI:** 10.3390/ijerph20010752

**Published:** 2022-12-31

**Authors:** Kellie-Ann Armitt, Janette Young, Rose Boucaut

**Affiliations:** Allied Health and Human Performance, University of South Australia, Adelaide, SA 5000, Australia

**Keywords:** ageing, human–animal interaction, human–animal bond, innovation, multi-species, health promotion, Ottawa Charter, companion animals, pets

## Abstract

This study explores the challenges facing a pilot project aiming to foster homeless cats in an Australian residential aged care facility. The global COVID-19 pandemic stalled the project but also presented an opportunity to gain reflective insights into the perceived barriers, enablers and tensions involved in seeking to implement pet animal inclusion in residential aged care. Perspectives from aged care management, animal welfare services and researchers/project managers were all sought using semi-structured interviews, and themes developed using a qualitative descriptive analysis. Perceived barriers to the project before and after the pandemic were not dissimilar with four key themes emerging: competing priorities, risk and safety, resources, and timing. All existed differently across stakeholder groups creating tensions to be negotiated. These themes are then mapped to the competencies established by the International Union of Health Promotion and Education (IUHPE) for undertaking health promotion, demonstrating that this skill base can be drawn on when seeking to implement human–animal inclusive projects. Creating supportive healthful environments for frail older persons is a moral imperative of extended lives. Health Promotion skills as outlined in the Ottawa Charter and IUHPE competencies for health promotion workers need to be extended to include animal services, agendas and cultures to promote multi-species health promotion into the future.

## 1. Introduction

Creating supportive, healthful environments for frail older persons is a moral imperative in societies blessed with ever-extending longevity. While many human beings share their domestic lives with other species, multi-species sharing is rare in residential aged-care settings. This lack of sharing contradicts the expressed wishes of many older people to continue to have close relations with other species as they age [1,2], and the evidence that cross-species relationships improve the quality of people’s lives, in particular supporting them as they move through the stressful transitions that come with ageing [3]. Promoting wellbeing and salutogenic health (the active promotion of wellness and wellbeing as versus pathogenic approaches that focus on the prevention of disease and disablement) in people’s lives as they age can be seen to fit with the remit of health promotion workers around the globe who ascribe to the standards established by the International Union of Health Promotion and Education (IUHPE) [4]. The IUHPE Core Competencies and Professional Standards have been adopted internationally, including in Australia, the location of this study for the purposes of individual registration and educational program accreditation. The competencies are built around the Ottawa Charter [5] and unpack the knowledge, skills and performance criteria needed to implement health promotion in communities and societies.

This paper discusses a pilot project focused on introducing pet animals into a communal residential aged care facility, with a specific aim to understand management issues in seeking to achieve this. The core results and thematic analysis presented are derived from interviews undertaken with key stakeholders in the project. The discussion turns to the intersection of these themes with the IUHPE Core Competencies as one of the first attempts to map these competencies to health promotion work undertaken within a cross-species milieu.

### 1.1. The Project

The Foster Cat Project was a joint venture between an animal rescue service, an aged care facility and university researchers. The original aim of the project was to place two cats at the facility full-time for a period of three months. The cats would live among the residents in a dedicated wing of the facility and would be cared for by a select group of staff and volunteers who had undergone specific training. The project aimed to develop a practical working model of companion animals, including both foster animals and personal pets, which could be replicated in other facilities in line with the Ottawa Charter action areas of creating health-promoting environments. The global health crisis of COVID-19 stopped the program in its tracks, but also presented an opportunity to gain unique insights into the effects of a pandemic on the implementation of healt-promoting innovation both prior to and at the onset of public health pandemic responses. Hence, the aim of this study was extended to include an understanding of how organizational factors, coupled with the unique challenges of a global pandemic impacted the proposed cat-fostering program.

The effects of the COVID-19 pandemic have been felt around the globe, but the negative impact has been disproportionately great for those in residential aged care. This population was subjected to harsh lockdown protocols to limit the spread of the virus, with some Australian facilities putting a complete ban on visitors, going above and beyond the government recommendations [6].

### 1.2. Australian Royal Commission into Aged Care

Prior to the pandemic, an Australian Royal Commission into Aged Care Quality and Safety was established in response to reports of neglect and substandard care in aged care facilities [7]. The aim of the Commission inquiry was to investigate the quality of aged care in Australia and to determine how these services could be improved. The interim findings described the industry as being characterized by a lack of innovation and not being built around the people it is supposed to help, rather, being focused on funding procedures and processes. The Commission called for real change [8]. A scoping review analysing innovative aged care models from around the globe was conducted as part of the inquiry; this review determined that models using a person-centred approach were more conducive to good quality care [9]. An important characteristic of this approach is allowing residents more choice in how they live, with respect to their individual history, values and experience [9]. For many older Australians, this includes living with a pet and maintaining opportunities to interact with animals [1], hence considering how to facilitate human–animal connections in aged care settings is indicated.

### 1.3. Animals and Ageing

The current rate of pet ownership in Australian households sits between 61 to 69% [10]; however, amongst those aged 70 years or older, pet ownership rates drop, with this being just 45% in the most recent age-identifying market research [11]. Data are scarce however the more recent figures estimate that around 6.75% of animals in Australian shelters are surrendered for reasons related to human ageing, such as the owner’s poor health or relocation to an aged care facility that is not pet friendly [12]. While there has been some headway made in this space with the recent development of policies aimed at integrating animals into aged care [13], the reality is that pet-friendly facilities remain limited. The number of facilities that will consider allowing residents to keep a pet range from 14% in Victoria and New South Wales, to 35% in Tasmania [12].

The therapeutic value of human–animal interaction is a popular study topic in the health care setting. A bibliometric analysis of human–animal interaction articles between 1982 and 2018 found there has been a steady increase in both the annual number of publications and the diversity of journals publishing these articles [14]. However, evidence in this field is evolving and is plagued by studies with methodological limitations and contradictory findings. A recent systematic review of the literature concerning human–animal interactions and older people highlighted the need for more rigorous research but concluded overall that there is real potential for animals to benefit the health and well-being of this population [15]. Given Australia’s ageing population, exploring this potential is important [16].

Research into the impacts of pets on older persons’ health largely falls into two areas of investigation: explorations as to the role of pets in community-dwelling older adults, or the impacts of visiting companion animals brought into health care facilities for short visits accompanied by a handler. Only one known research project (from the U.K.) has considered the experiences of living with one’s own personal pet in a communal aged-care facility [17].

The strongest bio-medical evidence to date supports a link between pet ownership and heart health. Friedman et al. [18] undertook a repeated-measures observational study with their findings showing that just being in the presence of their pet could reduce ambulatory blood pressure in mildly hypertensive people aged 50–83. Another observational study found that in a population of elderly hypertensive adults, those who were current or past pet owners had a reduced mortality rate and reduced incidence of a fatal cardiac event [19]. Gee and Mueller’s [15] systematic review identified that while pets could not be linked to lower rates of clinical mental health diagnoses, pet ownership could soften the edges of life transitions for older pet owners. The extent of this ‘softening’ can be extreme, however. For example, a recent small qualitative descriptive study exploring self-reported health impacts for pet owners aged over 60 found that almost a third indicated that their pets had been protective against suicide [20].

In terms of animal visitations to aged care, a meta-analysis of intervention studies involving such programs concluded that the visits resulted in a moderate reduction of depressive symptoms [21]. Four of these studies were conducted in residential aged-care facilities. A small-scale randomized controlled trial conducted in an assisted living facility introduced a visiting dog twice a week for 12 weeks for the intervention group [22]. The intervention group performed specific skills with the dog, compared to the control group who participated in a social interaction session and were asked to reminisce. Physical activity, participation in activities of daily living and depression scores all improved significantly in the intervention group over the three-month period.

One recent qualitative study explored the benefits of having a personal pet live in an aged care facility amongst residents [17]. The themes that emerged from the interviews were how pets increased motivation to live and stay active, provided a vital connection to individual’s past self and identity, gave owners a sense of purpose and feelings of usefulness and lastly, made the transition to institutionalised living much easier [17]. The study highlighted the potential of enhanced social, physical and psychological improvements through living with pets full-time in residential-aged settings.

In summary, while research in this area has some limitations, the potential to improve the physical and psychological health and wellbeing of older people by maintaining relationships with animals is well grounded in research.

### 1.4. Innovation in Healthcare

Implementing change within the health care sector is not easy, even under normal circumstances. Generic issues plaguing the industry have been identified including, staff shortages, high workloads and inadequate funding [23], all of which may be determining factors in deciding the fate of new initiatives. Health care facilities need to ensure a safe environment, for workers, residents and visitors, making an innovation involving animals, especially challenging. Animals pose a unique set of safety concerns to humans, such as risks of zoonoses, scratches and bites [24], as well as being potential trip hazards and increasing the likelihood of falls [25]. The welfare of the animals involved is also an important consideration. The inclusion of shelter animals, who may have a traumatic past, could inherently hold risk, as exposing them to new, unfamiliar environments and people may be stress-inducing [26]. Health care providers have an obligation to ensure that the animal’s care needs are being met, and that they (animals) are not being placed in a dangerous environment.

Change in the health care industry is not always welcomed and not every innovation is successful. The perceived legitimacy of the innovation is crucial to its uptake [27]. Some of the factors that facilitate the adoption and sustainability of innovation are: evidence of effectiveness, cost–benefits, addressing local unmet needs; reasonable, appropriate structural and procedural program elements [27]. Staff attitudes within the host organization, particularly management, are also important. It is well-documented that successful change requires commitment and strong support from all levels of the organization [28], and this attitude must be reinforced by leaders [29]. In health care shifts in tangible resources and/or policies may also be necessary [28]. In residential aged care specifically, current policies are a significant barrier to introducing any innovation involving animals, and do not reflect the importance that older Australians place on animals and how they add value to their lives [1].

Therefore, whilst the use of human–animal interaction in aged care has shown promising results, there is currently an evidence-practice gap as health care providers must weigh the possible risks against the perceived benefits. For those in residential care, a fostering program such as the one proposed may be highly appropriate as potentially it could still elicit the enhanced health benefits of living with a pet without the burden of personal responsibility for the animal [30]. The pilot action research project “Foster Cats” offered a chance to trial the safe incorporation of animals into an aged care facility and to develop a multi-species model that could be extended in the future. Homeless cats were more accessible for this model development than personal pets, and there was also the potential to address the number of rescue cats euthanized, identified as close to 20% by one of the largest animal rescue services in Australia in 2020–2021 [31].

The Foster Cat Project was initiated in discussions between the researchers and an animal welfare organization. Following a 12-month process of recruitment of a partner aged care facility, participatory engagement and collaboration with residents and staff of the facility to develop both the research (4 ethics applications) and applied project processes, the project was just about to commence when the global pandemic closed both the nursing home and the animal shelter and sent the researchers home to work indefinitely. While it had always been the intention to include management and policy analysis within the action research framework, the pandemic facilitated the opportunity to also focus on this unanticipated conundrum and link interview conversations to the unprecedented impact of a pandemic.

## 2. Methods

This study explored the perspectives of key organizational stakeholders in the cat-fostering project using a descriptive qualitative approach. A single case study design [32] facilitated a thorough in-depth analysis of the challenges faced by the Foster Cat Project.

Participants were purposefully selected who were information-rich and had firsthand knowledge [33] of the planned Foster Cat Project. Participants had varying levels of involvement with the program, ranging from high-level management within the participating organizations, to researchers with a passion for facilitating human–animal interactions in aged care. The research was approved by the human and animal research committees of the researchers’ university. A total of seven participants agreed to be included in the study. In keeping with the ethics approval, participants are not identified here by job title or place of employment to ensure anonymity across the three sectors of the relatively small state in which the research was undertaken. Participants were allocated gender-neutral pseudonyms as an extra layer of identity protection as there was an uneven gender spread within the sample (6 females, 1 male).

Semi-structured interviews were conducted, framed around the following questions:How did the project initially attract interest?How did organizational hierarchy/structure impact on the project?What were the main barriers prior to COVID-19?Were there additional/different barriers after COVID-19?How did systems/structures within the organization act as barriers?What are the possible long-term implications of COVID-19 on this and future innovations?

Interviews were conducted between January and June 2021 via Zoom by the first author and lasted 25–60 min with each participant only being interviewed once. Interviews were recorded and transcribed verbatim to ensure rigour, and transcripts were individually reviewed by the first and third authors as means of addressing ethical concerns at the second authors intimate engagement in the project. Codes were assigned consisting of key phrases and concepts regarding the challenges facing those involved with the Foster Cat Project and the impact of the COVID-19 pandemic. Multiple coders were used to improve the trustworthiness of the emergent themes [34] and inductive thematic analysis was used [35] to identify the common threads that were present across the entire set of interviews [36]. Themes were then discussed and refined, before being mapped visually to better interpret the results. Participants were provided with copies of their interview transcripts, as well as the resulting thematic charts, and asked to verify that their own perspectives were accurately captured to avoid misinterpretation of the data by the research team. No changes were requested.

## 3. Results

A total of seven interviews were conducted and this included representatives from the animal rescue organization (*n* = 3), the residential aged care facility (*n* = 2), and university researchers (*n* = 2), hence all major stakeholders in the project were encompassed. Findings were categorised into four themes: Competing priorities, Risk and safety, Resources and, Timing; with ten sub-themes identified. The themes and subthemes are identified in Table 1, with the grey shading indicating those themes impacted by COVID-19.

Each of the four themes was present both before and after the start of the pandemic, but with different nuances; seven subthemes were identified as being changed significantly by the pandemic.

### 3.1. Theme: Competing Priorities

While each participant recognized the potential benefits of the foster cat program, there were different priorities that impacted the decision making of key stakeholders.

#### 3.1.1. Intra-Organizational Priorities

While there had been energy and commitment to progressing the Foster Cat Project across 12 months prior to COVID-19, once pandemic restrictions were implemented across Australia the main priority for aged care facilities became keeping residents and staff safe from the virus at all costs. Concerns about insufficient Personal Protective Equipment (PPE) and adhering to infection control practices took precedence over all else, as highlighted by Robin:


*‘Additionally, all we were worrying about was, you know, where’s… where’s the next box of stuff [PPE] going to be coming from? That was just really what it was all about’.*


#### 3.1.2. Inter-Organizational Priorities

The integration of animals was identified by all participants as an opportunity for improvement in the delivery of aged care in Australia, as illustrated by Ashley:


*‘There’s so much more we can be doing with aged care and…certainly providing them with things more like home and the companionship of an animal, or the opportunity to interact with animals is really important’.*


Yet, this desire to improve the quality of life for residents and deliver person-cantered care is at odds with the need to protect the business interests of aged care facilities. Ashley also identified that having to accept liability in the event of a safety incident was a strong motivator when decisions about the program needed to be made:


*‘The other one…was the zoonotic diseases. Who was going to be responsible for the cost of the [additional] testing [of cats], and whose insurance did that sit on should there be an issue that arose?’.*


### 3.2. Theme: Risk and Safety

Risk and safety were a strong theme in both pre- and post-COVID-19 restrictions, but the source of the perceived risk changed once the pandemic began.

#### 3.2.1. Organizational

Each organization had a desire to protect the business from potential complaints or the threat of legal or financial repercussions should there be a negative outcome from the implementation of animal-inclusive innovation. This was illustrated by Jamie’s blunt statement:
*‘The fear of litigation is really strong’,*
unpacked by Robin’s comments:


*‘As the world has become more legalistic, access for resident’s families, for example, bringing pets in to visit and other residents being able to approach them becomes fraught with difficulty’.*


Managing human co-living environments, encompassing not only residents but also their visitors and families is complex and complicated by concerns of litigation.

#### 3.2.2. Animal to Human

Prior to COVID-19, the main concern was around the potential risks that cats might pose to residents. Planning for minimizing risks surrounding zoonotic diseases, bites and scratches slowed the progress of the program and was a consideration for all organizations.


*‘You’ve also got to ensure that you if you’re putting an animal into a potentially immunocompromised population that you’re not putting them [residents] at greater risk.’*
(Ashley)

Part of this was the element of the unknown that came with using rescue cats. Given that most cats surrendered to the animal shelter have an unknown medical and personal history, additional safety protocols such as enhanced screening and testing were deemed necessary to try and ensure resident safety, although this did put the program behind schedule.

Some participants believed these extra measures were overly cautious as they went above and beyond the usual processes for adopting an animal out of the shelter:


*‘But if you look at…a lot of the things they bring up are around, oh what if someone trips over an animal? What if somebody gets scratched or bitten?...Frail elderly people, you know…I mean they forget that frail, elderly people for hundreds of years have had animals in their homes.’*
(Lee)

Ironically once COVID-19 emerged animals no longer posed the greatest risk to residents, but rather other humans did, and this was the reason why facilities were placed under strict lockdown protocols.

#### 3.2.3. Human to Animal

Prior to COVID-19 restrictions, there were already concerns about the welfare of the animals going into the residential facility given that employees are paid to take care of people, not animals:


*‘It would then…become a responsibility for our most poorly funded resource which is our activity staff. And I mean you know, I mean of course that had implications for how well the animals were kept.’*
(Robin)


*‘The reality was there’s more than enough for staff to do without having to look after animals.’*
(Lee)

After COVID-19 restrictions were implemented, these concerns intensified, as staff became overloaded with additional responsibilities relating to the new protocols:


*‘Staff were completely overwhelmed by the additional responsibilities they had on top of their own jobs…there was a real risk that something like the animal’s well-being could have been detrimentally affected.’*
(Robin)

### 3.3. Theme: Resources

Access to financial, human and animal resources was a strong theme both before and after the start of the pandemic.

#### 3.3.1. Financial Resources

Prior to the pandemic, a significant barrier was deciding who would assume the cost of the animal testing, given that the foster program itself had very limited funding. There was also no guarantee that every cat tested would be deemed suitable to enter the facility, so costs could become overwhelming if multiple cats had to be tested before finding a feline candidate:


*‘Because the whole aim of this project was trying to put something in place that could be implemented and run long term, so if it was suddenly going to be very expensive, it was going to be very hard to justify how you will actually do it.’*
(Chris)


*‘Lifestyle initiatives, such as the foster cat program, have the least amount of funding in residential aged care making them difficult to implement and sustain: ‘And the, the reality is that, in our funding models, …lifestyle funding is probably about the most marginal [budget] line we have.’*
(Robin)

From an organizational point of view across both aged care and animal welfare entities, without specific funding to facilitate the innovation existing, already tightly squeezed resources needed to be utilized.

#### 3.3.2. Human Resources

Having the right people involved and fully committed at every organization was identified by every participant as being crucial to the initial uptake and then the sustainability of the foster cat program:


*‘You have to have full board and management buy-in, and it has to be strategic because if it’s not, it will sort of come and go and wax and wane.’*
(Sam)

This was similar to Jamie’s comments about how a personal interest in animals is important:


*‘You have to have someone who’s interested…in the idea and, you know, and has a sense of understanding that human–animal relationships exist and that they can be important’.*


Prior to COVID-19 restrictions, a significant challenge was staffing levels at the animal rescue service and the aged care facility and whether they could meet the demands of the project. This was overcome by recruiting volunteers for the program. However, once COVID-19 restrictions were in place, anyone deemed non-essential was denied access to the aged care facility due to lockdown protocols. On the animal rescue side of the project, there was a dissolution of employment positions dedicated to programs such as the Foster Cat Project in an effort to redistribute resources to manage the pandemic emergency:


*‘The other side of it is within shelters, a lot of positions that were for outside projects have obviously had to close down…like…those positions no longer exists [sic] because of restrictions COVID placed on it.’*
(Alex)

#### 3.3.3. Animal Resources

While prior to the pandemic discussions as to the accessibility of animals had focused on obtaining “the right cat[s]” for the project, this issue was overtaken in the pandemic by the global rush to acquire pets that lead to animals being adopted out by the general public at unprecedented rates [37]:


*‘Through the initial part of the pandemic, we, every weekend every single animal was adopted.’*
(Ashley)

Even if the project had been able to run in the aged care facility during COVID-19, there were no homeless cats to play their part in the project.

### 3.4. Theme: Timing

The timing of the foster cat program also coincided with several other influential events that were reported as posing barriers to the project’s implementation.

#### 3.4.1. Personnel Changes

The timing of personnel changes within one of the involved organizations was identified by three participants as being a barrier prior to the pandemic. The different perspectives that new personnel can bring were perceived to have slowed the program down:


*‘There were a lot of things that I guess the first manager didn’t really think of that she [new manager] did……And I think everything she was doing, while it slowed it down, we would have gotten there.’*
(Alex)

However, while this change slowed the project, it was felt that the project would have run had it not been for the overwhelming impact of COVID-19.

#### 3.4.2. The (Australian) Royal Commission into Aged Care

The timing of the program coinciding with the initial impact of the Royal Commission into Aged Care, and this was acknowledged as a major barrier by two participants. The intense scrutiny placed on the industry forced providers to become even more risk averse to improve public perceptions of delivery of care. Lee and Robin reported how this, coupled with the increased regulations because of COVID-19, made innovation in the aged care space almost impossible:


*‘If we could separate those two points and one had not occurred, [for example] the pandemic on its own…I think we’d be far more, far more enthusiastic about moving in directions that have some element of risk associated, you know, risk for people’s well-being.’*
(Robin)


*‘I think until, you know, we get the results and outcomes of the Royal Commission…I don’t think any organizations are going to be jumping in to be doing the ‘nice’ stuff. They’ll be too busy doing the ‘we have to’ stuff.’*
(Lee)

#### 3.4.3. COVID-19

The timing of the program coinciding with the pandemic was identified by all participants as the most important factor in the suspension of the project. Every participant stated that the other problems they had encountered were not insurmountable and would have eventually been resolved, but COVID-19 made the project impossible, as highlighted by Chris:


*‘Well, at the moment, it’s kind of ended up putting this project almost in the too hard basket. So it’s pretty much been decided that at the moment, it won’t run’.*


Sam made similar observations of how COVID-19 impacted the project: *‘Yes, I could see that COVID would have added that extra layer of risk that a risk-averse organization would have taken that opportunity to go well we’re not doing that anymore’*.

In summary, there were four major themes identified as impacting the implementation (or not) of an innovative human–animal relationships project in a communal living, residential aged care setting. These were: Competing priorities both between and within organisations, Risk concerns, encompassing organizations, humans and animals involved, Resources, financial, human and animal, and Timing including major external factors (a pandemic and a Royal Commission) and more routine changes such as personnel changes.

## 4. Discussion

In this discussion, the International Union for Health Promotion and Education (IUHPE) Core Competencies and Professional Standards for Health Promotion [4] are meshed with a discussion of the themes as a means of considering how future health promotion projects that seek to include pet animals in aged care can draw on these key skills for the benefit of both older people and animals. The core competencies have been adopted internationally for the purposes of individual registration and educational program accreditation and outline the knowledge, skills and performance criteria deemed necessary to implement health promotion as outlined in the Ottawa Charter [5]. There are 9 domains with 46 competency statements. This discussion seeks to extend our understandings of the implementation of these competencies beyond human-centric understandings of health promotion. Table 2 presents domains and competency statements that can be seen to have underpinned and established the Foster Cat Project.

Foundational to the project was the existing and growing research base that exists on the health-promoting nature of human–animal relationships with the aim being to further strengthen this knowledge base (Domain 9: Statement 9.4). Preparatory work encompassed advocating for both human and animal health through actively engaging and seeking to influence stakeholders (Domain 2: Statement 2.2). Change was enabled by bringing human and animal services together to discuss their respective species health concerns; a cross-species approach to reorienting services to promote health and reduce inequities (Domain 1: Statement 1.5). The application of these competencies led to the building of a successful foundational partnership across sectors aimed at promoting both human and animal health (Domain 3: Statement 3.3).

### 4.1. Connecting Themes to IUHPE Competencies

A range of IUHPE competencies can be identified as assisting future health promotion workers to recognise and respond to the themes and subthemes identified by participants. Table 3 summarises the connections made and referred to here. This is the first known attempt to connect the IUHPE competencies with human–animal practise, but the connections are merited given the evidence base of needs and benefits from human–animal focussed health promotion in the context of ageing.

While the COVID-19 pandemic added layers of complexity to the attempt at introducing foster cats into a residential aged care facility, the core themes identified as barriers faced by the project did not change per se. Rather the fundamental concerns were reshaped, reflecting a dramatically different environment impacting intimately on organization’s day-to-day operations. The capacity to read and assess the impacts of broader social, political and economic environments on health promotion actions is a core IUHPE competency (Domain 6: Assessment). In the case of the Foster Cat Project, staffing, cost and other organizational factors, while important, were not seen as insurmountable barriers prior to COVID-19. While they slowed the progress of the Foster Cat Project, appropriate solutions were being worked through when the global pandemic arrived in Australia and COVID-19 restrictions were put in place. Arguably, it was responses to the pandemic which made implementing the project impossible.

#### 4.1.1. Competing Priorities

IUHPE competencies require health promotion workers to use effective communication skills including being able to use culturally appropriate communication (Domain 4: Statement 4.3). Human versus animal focussed services present uniquely differing cultures, facing very different pressures. Health promotion workers need to seek out these differences in communication with each sector, then seek to communicate the differences in discussions with partners so that mutual understanding and respect for differences are facilitated.

Our research identifies the need to be aware of the impacts of highly visible political undertakings such as that of the Royal Commission into Aged Care on specific sectors. Prior to COVID-19, aged care stakeholders were open to suggestions and initiatives regarding having animal companions in a residential aged care facility. The Royal Commission had raised societal concerns voiced by the public about the way seniors in society are cared for. New aged care quality standards [38] including the requirements that aged care providers seek to actively enable older persons to continue engaging in activities of their own choosing, had provided a strong platform for the Foster Cat Project. While adding to the stress levels on aged care providers, prior to the pandemic the Royal Commission and the new standards had provided impetus for the project.

However, our findings highlight a range of paradoxical changes that COVID-19 brought about within the aged care industry during the crisis. The intense scrutiny on aged care and public media reporting of indignation at extreme enforced isolation and mounting death toll from COVID-19 in these settings placed service providers under enormous pressure to respond [39]. Crisis responses to COVID-19 led to aged care services being locked down for months on end, and the industry prioritised the preservation of life over the preservation of quality of life in its attempt to protect residents [39,40]. However, given the known links between social isolation and increased chances of morbidity [41], and the knowledge that institutionalised living fosters isolation and loneliness [42] ironically periods of lockdown served to exacerbate negative mental health impacts for many residents in residential aged care [39,40].

The skills of a larger accredited health promotion workforce could have been invaluable in the pandemic in this regard. Understandings drawn from the Ottawa Charter [5] of seeking to actively create health-promoting environments, rather than simply presume that medicalized health protection creates this could have been used to identify the risks in ‘merely’ preventing deaths [39,40]. Had aged care services routinely encompassed human–animal relationships prior to COVID, the intense stress experienced by many residents and their loved ones engendered through lack of face-to-face interaction and physical contact could have been reduced with animals filling some human needs for touch and physical contact [43].

#### 4.1.2. Risk and Safety

Our study highlights the way in which foundational concerns of risk differed across human and animal-focused stakeholders. For animal rescue services primary concerns were ensuring that animals were kept safe and that there were no negative occurrences that could impact their mission of animal rescue; for example, zoonoses caused by the inclusion of animals with an unknown health history in a setting with frail immunocompromised persons. Aged care services, while conscious of the risks of zoonoses and injuries to humans, were more concerned with human resource management (overwhelming already over-stretched staff), and potential litigation should unmanaged harm (to a person) occur. Again, the need for the IUHPE competencies of Communication (Domain 4) and Assessment (Domain 6) skills in furthering human–animal inclusive health promotion models is indicated in communicating the shared but nuanced differences between sectors.

Findings from the current study strengthen the existing evidence surrounding the challenges of organizational change. A systematic review [44] identified that support from key stakeholders, as well as adequate funding and resources were strong influences on the implementation and sustainability of health care innovations. Wavering commitment across organizations and limited resources were identified as significant barriers by all participants in the present study, both before and after COVID-19.

#### 4.1.3. Resources

Awareness of the lack of additional resources within both aged care and animal welfare was a consistent dynamic as the project developed. However, the far more nebulous, but powerful resource of energy and enthusiasm was high. As noted by several participants the energy generated by animal lovers was key to the project almost succeeding. The suggestion is that in order to succeed human–animal projects inherently need animal lovers, and this of itself is a resource that needs to be overtly identified in the process of resources assessment and analysis as per the IUHPE competencies (Domain 6: Statement 5).

One of the starkest findings in this human–animal focused research was the impact on animal resources in the pandemic. From being built around the superfluity of homeless cats, COVID-19 led to the scenario outlined by Ashley of the shelter emptying of cats during the pandemic. This indicates the need to be not only identifying existing resources (Domain 6: Statement 6.5) but also monitoring them, particularly in times of unprecedented change. In addition, COVID-19 led to changes in the allocation of human resources within the animal rescue service as human resources that had been focused on the “luxury” of engaging with external stakeholders, were redirected back inwards to basic animal care services. Here, the intersection between the themes of Competing intra organisational priorities and Resources was manifest in the animal sector. It may have been possible to “keep back” several cats for the project, but the pressure on human resources in regard to priorities of caring for homeless animals, and finding fur-ever homes for them meant that there was no staff allocation available to pursue innovation in the pandemic. Aged care resources became more stretched with COVID-19 but perhaps could have accommodated cats due to the enthusiasm and commitment of staff. However, this is unknown due to the fundamental animal resources environment changes.

#### 4.1.4. Timing

The theme of Timing clearly is an overarching factor in the scenario explored by this research. Starting 12 months sooner we could perhaps be reporting on the impact of COVID-19 on an active Foster Cats Project in aged care! However, this was not the case, and the pandemic coincided with more common project implementation factors such as changes in key personnel. Such change is always a risk factor in facilitating health-promoting projects and forms part of the organizational understandings and knowledge base that good health-promotion assessment practice requires (Domain 6).

Understandings of Risk management at both organisational and societal (e.g., Royal Commission) levels changed with the timing of the pandemic. Changes to personnel can (and did) impact timeframes and timelines and deadlines needed to be reworked and renegotiated. Prior to the pandemic, the Royal Commission presented opportunities for the Foster Cat Project as an innovative, evidence-based way of responding to the new quality standards. Once pandemic restrictions were implemented what had looked like an opportunity became an impossible burden.

## 5. Conclusions

The evidence that cross-species relationships are vital for the health and wellbeing of many people means that health promoters with an interest in enhancing the health of frail older persons have a responsibility to engage in pursuing shared species opportunities in residential aged care settings. COVID-19 has brought about global change in the way we view animals and their role in our lives, while also creating unique forms of isolation, perhaps especially for the institutionalised elderly. The findings from this research, coupled with other research on the effects of lockdowns in aged care facilities, can be used to inform future health promotion work developing programs and policies that guarantee residents’ quality of life is better preserved during future lockdown responses. Ensuring pet animals are embedded into aged care is one (evidence-based) means of enabling pathogen focussed health protection responses to be more health-promoting. Both routinely, and in the face of future pandemics.

## Figures and Tables

**Table 1 ijerph-20-00752-t001:** Themes and Sub-themes *.

Themes	Sub-Themes
3.1. Competing Priorities	3.1.1. Intra-Organizational
	3.1.2. Inter-Organizational
3.2. Risk & Safety	3.2.1. Organizational
	3.2.2. Animal to Human
	3.2.3. Human to Animal
3.3. Resources	3.3.1. Financial
	3.3.2. Human
	3.3.3. Animal
3.4. Timing	3.4.1. Personnel changes
	3.4.2. Royal Commission
	3.4.3. COVID-19

* Numbered as per text; grey = those changed by COVID-19.

**Table 2 ijerph-20-00752-t002:** IUHPE Core competency domains and statements underpinning & establishing the Foster Cat Project.

IUHPE Core Competency Domain	Core Competency Statement	Role in Project
9. Evaluation & Research	9.4 Use research and evidence-based strategies to inform practice.	Foundational understandings
2. Advocate for Health	2.2 Engage with and influence key stakeholders to develop and sustain Health Promotion action.	Preparatory
1. Enable Change	1.5 Work in collaboration with key stakeholders to reorient health and other services to promote health and reduce health inequities.	Preparatory
3. Mediate through partnership	3.3 Build successful partnership through collaborative working, mediating between different sectoral interests	Outcome

**Table 3 ijerph-20-00752-t003:** Research themes & links to IUHPE competency domains and statements.

Themes	Competency Domain Indicated	Competency Statement
Competing Priorities	4. Communication	4.3 Use culturally appropriate communication methods and techniques…
Risk & Safety	4. Communication	4.3 Use culturally appropriate communication methods and techniques… 6.6 Use culturally and ethically appropriate assessment 6.7 Identify priorities for health promotion action in partnership with stakeholders…
	6. Assessment	
Resources	6. Assessment	6.5 Identify the existing assets and resources relevant to health promotion action
Timing	6. Assessment	6.3 Collect, review and appraise relevant data, information and literature to inform health promotion action.

## Data Availability

It is not possible to provide public access to the raw data for this project. This is because the small size of the state in which this research was undertaken, with small highly networked organisational players, meant that multiple steps had to be taken to maximise participant anonymity/privacy; participant agreement to providing public access to even de-identified data was not sought. The authors are more than happy to discuss the research further however should readers wish to contact them and this can be informed by their continued access to the deidentified data for the time specified in our ethics approval (5 years).

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
