# Peer review of "A Qualitative Analysis of Management Perspectives on Seeking to Implement the Foster Cat Project in Residential Aged Care in the Context of COVID-19"

_ijerph, 2022, doi:10.3390/ijerph20010752_

Round 1

Reviewer 1 Report

Congratulations on your work. I find it very interesting and with a great practical application that will greatly improve the quality of life of an important part of the population. Thank you for your input.

With an interest in improving, in my humble opinion, the quality of your work, I recommend the following modifications:

* The title should contain the type of study (qualitative) and should include the project on which the work is based and be a little more descriptive of the project

* The abstract does not include what is later found in the sections of the study. I would appreciate a little more fidelity in your wording and make it a little more attractive.

*The second paragraph of the introduction (lines 46-50) seems to refer to the current study, so it should be in the results or in the discussion of this work.

* Do not incorporate the explanation for the suspension of the first part of the study during the COVID pandemic. Simply report what has been done on the dates you have neglected.

* Some parts of the introduction can be used for discussion, leaving this section easier to follow. There is no need for so much justification.

* In the discussion section, especially in the first part, you are mixing information that should be in the results section (tables 2 and 3, for example). Leave all this new information that you have obtained from the interviews in the results and dedicate the discussion to compare it with the existing literature and to propose measures/solutions/proposals.

Author Response

Dear Reviewer 1

Many thanks for reviewing our paper and your helpful guidance.

Below we provide our responses to your specific points:

Reviewer 1

Author responses

Congratulations on your work. I find it very interesting and with a great practical application that will greatly improve the quality of life of an important part of the population. Thank you for your input.

With an interest in improving, in my humble opinion, the quality of your work, I recommend the following modifications:

Thank-you for your kind words.

* The title should contain the type of study (qualitative) and should include the project on which the work is based and be a little more descriptive of the project

The title has been reworked to be more descriptive of the paper’s focus and include the project methodology, and Foster Cat terminology

* The abstract does not include what is later found in the sections of the study. I would appreciate a little more fidelity in your wording and make it a little more attractive.

The abstract has been reworked to better reflect the content of the paper and is now at 207 words

*The second paragraph of the introduction (lines 46-50) seems to refer to the current study, so it should be in the results or in the discussion of this work.

With respect to our reviewers opinion, we have left this paragraph in situ as it is a sign-posting paragraph for the reader

* Do not incorporate the explanation for the suspension of the first part of the study during the COVID pandemic. Simply report what has been done on the dates you have neglected.

We have assumed that this is referring to (roughly) to lines 188-197.

As we straddle pre and during the pandemic across all the analyses (thematic and re IUHPE comps) we feel that this paragraph provides a timeframe snapshot. And continues the process of context setting that we see the background as providing.

* Some parts of the introduction can be used for discussion, leaving this section easier to follow. There is no need for so much justification.

Thank-you for this suggestion. We gave thought to reapproaching our structure, however as the other reviewer thought that the project was well justified and the introduction was providing a sound overview of the topic (as we had intended) we have decided to leave as is. However as per the second reviewers suggestions we have removed some clutter in the text by removing numbers of participants noted in studies.

Reviewer 2 Report

The study is written very well, english is adequate and I did not find major language issues. Please note that all of the suggestions reported below regard small details and should not be considered as an absolute hindrance for consideration by the IJERPH editors.

  • I suggest to use capitals for the project name (Foster Cat Project) or identify an acronym
  • I suggest to remove the number of subjects/included studies of cited research from the text (e.g., see line 126)
  • Introduction gives a complete overview of the topic and of the intervention
  • Materials and methods section is adequate
  • Line 470: please use COVID-19 instead of Covid
  • Discussion and conclusion are adequate.

Author Response

Dear Reviewer

Thank-you for reading our paper and your helpful advice. Please see our response to your specific points below:

Reviewer 2

The study is written very well, english is adequate and I did not find major language issues. Please note that all of the suggestions reported below regard small details and should not be considered as an absolute hindrance for consideration by the IJERPH editors.

Thank-you

I suggest to use capitals for the project name (Foster Cat Project) or identify an acronym

Thank you – this has been done throughout the paper

I suggest to remove the number of subjects/included studies of cited research from the text (e.g., see line 126)

Thank-you for this suggestion and we have taken these out leaving only the numbers of participants and where they were from in the text.

Introduction gives a complete overview of the topic and of the intervention

Thank-you

Materials and methods section is adequate

Thank-you

Line 470: please use COVID-19 instead of Covid

Amended

Discussion and conclusion are adequate.

Thank-you
